# Effects of Marginal Zn Excess and Thiamine Deficiency on Microglial N9 Cell Metabolism and Their Interactions with Septal SN56 Cholinergic Cells

**DOI:** 10.3390/ijms24054465

**Published:** 2023-02-24

**Authors:** Anna Ronowska, Agnieszka Jankowska-Kulawy, Sylwia Gul-Hinc, Marlena Zyśk, Anna Michno, Andrzej Szutowicz

**Affiliations:** 1Department of Laboratory Medicine, Medical University of Gdansk, 80-210 Gdansk, Poland; 2Department of Molecular Medicine, Medical University of Gdansk, 80-210 Gdansk, Poland

**Keywords:** acetyl-CoA, mitochondrial metabolism, N9 microglia, neurodegenerative disease, SN56 neuronal cells, thiamine deficiency, Zn excess

## Abstract

Mild thiamine deficiency aggravates Zn accumulation in cholinergic neurons. It leads to the augmentation of Zn toxicity by its interaction with the enzymes of energy metabolism. Within this study, we tested the effect of Zn on microglial cells cultivated in a thiamine-deficient medium, containing 0.003 mmol/L of thiamine vs. 0.009 mmol/L in a control medium. In such conditions, a subtoxic 0.10 mmol/L Zn concentration caused non-significant alterations in the survival and energy metabolism of N9 microglial cells. Both activities of the tricarboxylic acid cycle and the acetyl-CoA level were not decreased in these culture conditions. Amprolium augmented thiamine pyrophosphate deficits in N9 cells. This led to an increase in the intracellular accumulation of free Zn and partially aggravated its toxicity. There was differential sensitivity of neuronal and glial cells to thiamine-deficiency–Zn-evoked toxicity. The co-culture of neuronal SN56 with microglial N9 cells reduced the thiamine-deficiency–Zn-evoked inhibition of acetyl-CoA metabolism and restored the viability of the former. The differential sensitivity of SN56 and N9 cells to borderline thiamine deficiency combined with marginal Zn excess may result from the strong inhibition of pyruvate dehydrogenase in neuronal cells and no inhibition of this enzyme in the glial ones. Therefore, ThDP supplementation can make any brain cell more resistant to Zn excess.

## 1. Introduction

The energy homeostasis of the brain is a very complex process due to the restrictions of substrate transport through the blood–brain barrier against high-energy requirements. It results in the susceptibility of neurons to exogenous metabolic stress, also facing limited glycogen storage in astroglial cells. In fact, the brain uses about 20% of plasma glucose, but it constitutes only 2% of the total body weight [1]. Moreover, most of the brain glucose supply is metabolized by neuronal cells [2]. Energy deficits are observed in the brains of patients with several neurodegenerative diseases such as Alzheimer’s disease (AD), Parkinson’s disease, mild cognitive impairment and diverse animal transgenic models of neurodegeneration [3,4,5,6]. The most important source of energy deficits in the brain is the suppression of the pyruvate dehydrogenase complex (PDHC), tricarboxylic acid (TCA) cycle enzymes and respiratory chain complexes. Our past studies showed that different neurotoxins associated with AD pathomechanisms, such as NO or Zn excess and amyloid-β and thiamine diphosphate deficits (ThDP), directly inhibited the activities of the PDHC and other mitochondrial enzymes: aconitase, NADP-isocitrate dehydrogenase (IDHC-NADP) and the 2-oxoglutarate dehydrogenase complex (OGDHC) [5,7]. This inhibition suppressed the utilization of acetyl-CoA in the TCA cycle and the respiratory chain yielding ATP deficits in mitochondria [7,8,9,10]. The reduction in acetyl-CoA synthesis is evidenced in the cellular and animal models of cholinergic encephalopathies [7]. On the other hand, the inhibition of acetyl-CoA synthesis in cholinergic neurons led to decreases in acetylcholine synthesis and release. This caused a functional and structural impairment of cholinergic neurons yielding an impairment of their functions [8,9,10].

Therefore, the loss of PDHC activity and energy hypometabolism are hallmarks of diverse brain encephalopathies. One cause of these conditions is thiamine diphosphate (ThDP) deficit. ThDP is a cofactor of many enzymes that play an important role in key metabolic pathways: the TCA and pentose phosphate pathways. The E1 subunit of the PDHC and OGDHC is activated by ThDP and also the branched-chain alpha-keto acid dehydrogenase complex (BCDHC), 2-oxoadipate dehydrogenase complex (OADHC), transketolase and ThDP-dependent α-oxidation enzymes [11]. It is caused by thiamine deficiency (TD) resulting from a chronically low-thiamine diet, starvation, alcoholism, mental or physical disabilities or some chronic extensive medical procedures. TD may clinically manifest as angiopathy, myopathy or encephalopathy, which may be a life-threatening condition. However, most TD cases remain at a subclinical level. Nevertheless, it makes individuals more susceptible to other pathogenic conditions such as AD [12]. One of the neurodegeneration-inducing factors may be the TD-evoked activation of microglia toward a toxic M1 phenotype, demonstrated by several-fold increases in the expression of specific CD11B and CD68 antigens [13]. The alterations appear quickly preceding those taking place in neurons [12,14]. M1 microglia release several pro-inflammatory cytokines, such as IL-1β, IL-6, TNF-α and INF-γ, which may aggravate the metabolic and functional dysfunction of adjacent ThDP-deficient neurons and oligodendroglial cells [15]. However, there are data on the existence of the parallel cytoprotective activities of microglia, which are linked with the release of anti-inflammatory and neuroprotective factors such as IL-8, IL-10, BDNF or NGF [16,17]. However, there is no data on whether similar shifts are likely to take place in ThDP-deficient microglial cells.

Moreover, TD-evoked energy deficits are linked with the prolonged depolarization of glutamatergic neurons containing Zn in their terminals [8,18,19]. Zinc is the most abundant cation in the human body. Zn concentration in human plasma is 0.008 mmol/L, and in CSF, it is 0.0018 mmol/L [20]. This cation is mainly located in the gray matter, where its intracellular concentration is about 0.15 mmol/L. Most of all Zn is in the synaptic vesicles of glutaminergic terminals where its concentration is about 1 mmol/L and it forms weak complexes with L-glutamate [20]. However, 80% of the cation is bound with proteins, and therefore, the concentration of free Zn is in the nmol/L. Glutamatergic neurons form about 50% of synapses in the brain. In physiologic conditions, Zn is released to synaptic clefts where it is quickly cleared mainly by astrocytes and postsynaptic neurons. However, during extensive depolarization, Zn is co-released with glutamate in excess, reaching concentrations of over 0.30 mmol/L. It provides a higher accumulation of Zn in postsynaptic neurons and glial cells yielding excitotoxic injury [8]. Then, the ions are transported to mitochondria affecting energy metabolism [21,22]. There are data showing that the physiological level of free Zn in the matrix of mitochondria is lower than 3 pmol/L [23]. Thus, aberrant zinc homeostasis may be an additional pathogenic factor in diverse neurodegenerative conditions [24,25]. Our recent findings indicate that an increased zinc concentration caused the inhibition of the activities of the PDHC, OGDHC, aconitase and NADP-isocitrate dehydrogenase and a decrease in acetyl-CoA in differentiated cholinergic SN56 neurons derived from a mouse septum [26,27]. Low ThDP levels caused the augmentation of Zn cholinotoxicity by promoting intraneuronal Zn accumulation [9]. On the other hand, astroglial cells that are able to take up Zn from the extracellular space exerted a cytoprotective effect on the neuronal cells [8]. Moreover, Zn in relatively low 30–50 µmol/L concentrations promoted amyloid-β aggregation leading to excessive microglia activation promoting the release of pro-inflammatory cytokines [28,29]. These compounds would augment the simultaneous release of Zn and glutamate from glutamatergic nerve terminals evoking the vicious cycle of neurodegeneration [1]. However, there is no data on whether microglia might take up Zn from the adjacent synaptic extracellular space. Each of those properties would implicate the neuroprotective properties of microglia against Zn insults. There is also no experimental evidence on the putative interactions of thiamine deficiency with any Zn excess on their joint effects on the energy metabolism and function of microglial cells. Therefore, in the presented study, we investigate the alterations of the enzymes of energy metabolism in microglia taking place in the presence of those specific neurodegenerative signals, which may be the answer to how borderline neurotoxic signals may affect the interaction of microglia with cholinergic neuronal cells. Our data demonstrate that it might be the case.

## 2. Results

### 2.1. Concentration-Dependent Effect of Amprolium in Microglial N9 and SN56 Cholinergic Neuronal Cells

#### 2.1.1. Concentration-Dependent Effects of Amprolium on Thiamine Pyrophosphate (ThDP) Intracellular Content and Viability of the Cells 

The cells were cultured in the Minimum Essential Medium Eagle (MEM) medium containing 0.003 mmol/L of thiamine according to the manufacturer’s protocol [30]. Its use mimics the conditions of thiamine deficiency (TD). The recommended thiamine concentration in the culture medium is 0.009 mmol/L [30,31]. To achieve a control medium, we added 0.009 mmol/L of thiamine to the MEM. The control levels of ThDP in the N9 cells cultured in the medium supplemented with 0.009 mmol/L thiamine and in the thiamine-deficient medium were equal to 189.2 and 150 pmol/mg of protein, respectively (Figure 1a). The ThDP contents in neuronal SN56 cells in corresponding conditions were over two times lower, being equal to 92 and 70 pmol/mg of protein, respectively (Figure 1a). The use of a TD medium caused significant, although not high, decreases in ThDP in both cell groups (Figure 1a). Therefore, to aggravate intracellular ThDP deficiency, amprolium, a competitive inhibitor of thiamine transport, was added [32]. The addition of amprolium of up to 10 mmol/L to the medium supplemented with thiamine did not alter intracellular ThDP levels after 24 h exposure (Figure 1a). Moreover, the total cell number and viability of N9 and SN56 cells were not changed (Figure 1b–d). However, in the thiamine-deficient medium, amprolium caused concentration-dependent decreases in intracellular ThDP content and the number of N9 cells (Figure 1b). There was an 80% decrease in intracellular ThDP content in N9 caused by 10 mmol/L amprolium (Figure 1a). In the same conditions, the total N9 cell number was limited only by 38% and viability by 5% (Figure 1b–d). Thus, amprolium appeared to be a relatively strong suppressor of the ThDP level in N9 cells with IC_50_ equal to 2.34 mmol/L (Figure 1a) but only in the TD medium. However, the viability of N9 cells presented significant resistance to ThDP depletion (Figure 1c).

On the other hand, in SN56 cells, 5 mmol/L amprolium in the TD medium caused a small 20% decrease in ThDP content but a relatively high 35% decrease in SN56 cell count and a 12% increase in TB-positive cells (Figure 1a–d). These detrimental effects in SN56 were increased to 50% and 25% against controls at a 10 mmol/L amprolium concentration, respectively (Figure 1b,c).

#### 2.1.2. Concentration-Dependent Effects of Amprolium on Enzymes of Energy Metabolism and Acetyl-CoA Level in N9 Cells

Amprolium-evoked decrease in ThDP content in N9 cells was accompanied by concentration-dependent decreases in PDHC and aconitase activities. At a 10 mmol/L amprolium concentration, their activities were inhibited by 65% and 40%, respectively (Figure 2a). IC_50_ values for amprolium against PDHC and aconitase activities in N9, calculated from Dixon’s plots, were equal to 3.8 and 12.5 mmol/L, respectively (Figure 2a insert). In addition, 10 mmol/L amprolium decreased acetyl-CoA levels in N9 cells by 30%, with IC_50_ 18.8 mmol/L (Figure 2b). The decrease in the acetyl-CoA level in N9 correlated weakly with the inhibition of PDHC activity (r = 0.881, *p* = 0.048) (Figure 2c). On the other hand, in SN56 neurons, a relatively small amprolium-evoked decrease in PDHC activity was strongly correlated (r = 0951, *p* = 0.0037) with a profound depletion of acetyl-CoA content. These alterations indicate that cholinergic neuronal cells appeared to be 16 times more sensitive to intracellular ThDP deficits than microglial ones, comparing the slope values of their acetyl-CoA vs. PDHC correlation plots (Figure 2c). Citrate synthase and ICDH-NADP activities in N9 were not affected by intracellular ThDP shortages (Figure 2d).

SN56 neuronal cells grown in the TD medium were more susceptible even to small amprolium-aggravated intracellular ThDP deficits than N9 microglial cells (Table 1). The amprolium IC_50_ value against PDHC activity in SN56 was 17.5 mmol/L, and it was over four times higher than in N9 (3.8 mmol/L) (Figure 2a). On the other hand, IC_50_ against the acetyl-CoA level in neuronal cells was four times lower than in N9 microglia (Figure 2b), which is compatible with the greater susceptibility of the former to any cytotoxic insult [9].

### 2.2. Effects of Zn Excess on Viability Parameters of N9 and SN56 Cells, Depending on Thiamine Availability

#### Effects of Zn on Microglial N9 Cells

It has been shown that only free Zn cations in the extracellular compartment may display cytotoxic activity [22,26]. These data revealed that a 10% FBS, present in culture media, containing an optimal concentration of thiamine, complexed extracellular Zn up to 0.10 mmol/L which prevented cytotoxic influences in SN56 cells [16,17]. That was a maximal Zn concentration saturating FBS-binding capacity [22,26] (Table 1). Here, we demonstrate that N9 microglial cells, grown either in thiamine-supplemented or in TD media, displayed no susceptibility to borderline 0.10 mmol/L Zn concentrations (Table 1). The proliferating capacity and functional viability measured by cell counts were retained, and the trypan blue exclusion rate was not changed, and neither were the metabolic parameters (Table 1). The increase in extracellular Zn to 0.15 mmol/L in the thiamine-deficient medium caused a 55% and 71% inhibition of the N9 cell number and proliferation index but only a slight 5% loss of viability in the remaining cells, respectively (not shown). No significant alterations in viability and growth rate were found in N9 cultivated in the thiamine-supplemented medium, despite a 146% increase in Zn accumulation and a 54% inhibition of aconitase activity (Table 1). These data indicate that microglial cells may be relatively resistant to increases in extracellular/intracellular Zn, taking place under different pathological conditions generating the excitotoxic activation of glutamatergic neurons in the damaged brain.

Thus, 0.10 mmol/L Zn in the TD medium inhibited the growth rate and viability of SN56 by 56 and 53%, respectively. Despite a moderate 50% increase in intracellular accumulation, Zn inhibited SN56 PDHC, aconitase and ICDH-NADP activities by 56, 67 and 85%, respectively (Table 1). It caused a 76% drop in the proliferation index corresponding to a 56% decrease in cell count with a 41% fraction of nonviable neuronal, trypan-blue-retaining cells. In TD conditions, 0.15 mmol/L Zn caused a massive death of 75% of SN56 at the proliferation index of 0.2 and 62% fraction of nonviable cells. These data indicate that Zn accumulation does not depend on ThDP status in both cell types. However, intracellular Zn cytotoxicity strongly depends on the ThDP level in neuronal cells and much less on its status in the microglial ones. Therefore, for the studies of combined cytotoxicity, 5 mmol/L amprolium and 0.1 mmol/L Zn were selected as they exerted none or moderate negative effects on N9 microglial and SN56 neuronal cells, respectively.

### 2.3. Combined Effects of Zn Excess and Thiamine Deficits on SN56 Cholinergic Neurons Co-cultured with N9 Microglial Cells

In the TD medium, in control conditions, the co-culture of SN56 cholinergic cells with N9 microglia seeded on permeable inserts (see Section 4) resulted in no alterations in their ThDP content, growth index, viability (Figure 3a,c,d), PDHC, aconitase and ICDH-NADP activities (Figure 4a,c,d) as well as the acetyl-CoA level (Figure 4b). 

However, microglial inserts decreased the basal Zn level in SN56 by 45%, below the control value (Figure 3b). The combined treatment of SN56 with Zn and amprolium caused a semi-additive 60% suppression of the ThDP level, which was partially augmented by N9 inserts (Figure 3a). N9 inserts did not alter the final SN56 count and viability in control conditions. In the presence of amprolium + Zn in the medium, inserts only marginally reversed the suppression of growth but markedly reduced the loss of viability in the remaining SN56 from about 60 to 20% (Figure 3c,d).

The combined addition of 0.10 mmol/L Zn and 5.0 mmol/L amprolium caused an additive inhibition of the PDHC and aconitase in SN56 and N9 cells. Note that PDHC activity was inhibited by 100% in the presence of both inhibitors (Figure 4a). On the other hand, ICDH-NADP activity in SN56 was 90% inhibited, whereas in N9, it remained resistant to these conditions (Figure 4d). Inserts of N9 alleviated, in part, the suppressive effects of Zn + amprolium on SN56 neurons (Figure 3 and Figure 4). 

### 2.4. Interleukin 6 and Tumor Necrosis Factor-α Release

For the assessment of N9 functional viability, lipopolysaccharide 0.1 mmol/L was added in a duplicate validating experiment. This preparation resulted in 70-fold and 825-fold increases in IL-6 and TNF-α levels in the N9 culture medium, respectively (Figure 5a).

Concentrations of IL-6 released to the medium during the 24 h culture of SN56 alone remained at similar levels irrespective of experimental conditions (Figure 6a). Moreover, N9 inserts did not significantly increase IL-6 release. On the other hand, neuronal cells alone released very low amounts of TNF-α, and the additions of amprolium or Zn suppressed them to non-measurable levels (Figure 6b). The addition of N9 inserts to controls resulted in a 10-fold increase in accumulated TNF-α, which in turn was reduced by 70% upon the addition of amprolium with Zn but remained higher than in SN56 alone treated with amprolium + Zn (Figure 6b).

## 3. Discussion

There is a number of experimental data that present separate effects of either Zn excess or TD on the energy metabolism and function of brain cells [12,26]. However, these pathogenic signals may apparently combine in in vivo conditions [1]. Our recent studies revealed that in cholinergic neuronal cells, the effects of marginal Zn excess may be aggravated by coexisting borderline ThDP deficiency and alleviated by co-culture with astroglial C6 cells [8,9]. It was due to the fact that astroglial cells were able to accumulate the Zn excess [8]. However, the putative contribution of adjacent microglia to the above-mentioned neuroprotective mechanism was much less investigated [33,34]. Therefore, our previous studies justify the presented work on interactions between cholinergic neuronal and microglial cells in conditions reflecting the excitotoxic environment in TD brains. In such conditions, a low metabolic flow through a ThDP-devoid PDHC generates less acetyl-CoA, yielding ATP deficits and the prolonged depolarization of glutamatergic terminals prevalent in the brain [35]. This generates an excess of glutamate and Zn in synaptic clefts. Zn is cleared from this compartment by postsynaptic and peri-synaptic cells. Such activity was found in C6 and primary astroglial cells that in excitotoxic conditions took up Zn from the extracellular space reducing the overload of co-cultured neuronal cells with this cation [36].

The presented data demonstrate that culture in the TD medium resulted in a mild 20–25% drop in ThDP in microglial and SN56 cells, which is compatible with clinical data showing no overt TD symptoms in humans and experimental animals with 20% decreases in ThDP levels in their blood or tissues (Figure 1) [36]. On the other hand, 30–50% decreases in ThDP levels in an animal’s brain or human serum were accompanied by overt clinical symptoms including neurological and cognitive disturbances [37]. Moreover, here in in vitro conditions, an additional factor, amprolium, the inhibitor of thiamine transport, aggravated mild ThDP deficits in microglial and neuronal cells and suppressed acetyl-CoA metabolism yielding their increased mortality (Figure 1 and Figure 2). It indicates that conditions employed here might decently reflect those taking place in thiamine-deficient brains in vivo.

The PDHC activity and ThDP level in N9 microglia were 40–100% higher than those in SN56 neurons, both in thiamine-saturated and TD conditions (Figure 1a and Figure 2a). On the other hand, PDHC activity in SN56 was more resistant to TD than in N9, as demonstrated by IC_50_ for amprolium equal to 3.8 and 17.8 mmol/L, respectively (Figure 2a). They are compatible with the relatively greater decreases in ThDP in N9 against SN56, in response to increasing amprolium concentrations in the thiamine-deficient medium (Figure 1a). However, acetyl-CoA levels in control TD SN56 were over two times higher than in N9. Despite the relatively weak PDHC inhibition caused by ThDP deficit, it generated 16 times stronger suppression of acetyl-CoA content in SN56 than in N9 cells as calculated from acetyl-CoA vs. PDHC correlation slopes equal to 38.2 and 2.4, respectively (Figure 2b,c). However, in the thiamine-supplemented culture, SN56 neuronal cells displayed 140 and 35% higher PDHC activity and acetyl-CoA levels than those in the TD medium with a mild ThDP deficit, respectively (Figure 2a,b) [32]. Such shifts may be explained by the TD-dependent decrease in acetyl-CoA utilization in the mitochondrial TCA cycle and NAA synthesis, as well as for cytoplasmic ACh synthesis and lipid membrane recirculation, linked with impaired cholinergic neurotransmission [22]. On the other hand, in N9 microglia, TD caused no change in PDHC activity but caused a 30% drop in acetyl-CoA. It might be due to the increase in acetyl-CoA utilization for multiple acetylations during microglia activation under pathogenic conditions (Figure 2d) [38]. Nevertheless, relatively stable acetyl-CoA levels in microglia were possibly linked with the self-protective mechanisms against the increasing levels of pro-inflammatory cytokines (Figure 2b,c, Figure 6b and Figure 7) [1]. In addition, microglial cells have seven times lower activity of ATP-citrate lyase than neurons. It suggests that this enzyme limits the citrate-mediated withdrawal of acetyl units from the mitochondrial compartment [38]. 

The activated microglia are considered to be a characteristic feature of TD brains. It is known that activated microglia, just like macrophages, have high energy requirements for glucose and lactate [39]. Therefore, toxicity-resistant cells including lymphocytes and microglia are characterized by the content of high intracellular ThDP levels [40] (Figure 1). On the other hand, relatively low levels of ThDP in cholinergic neurons may be an additional factor impeding the supply of acetyl-CoA and energy through the PDHC, making them highly susceptible to these conditions (Figure 1b,d and Figure 3a,c, Table 1). No changes in aconitase and ICDH-NADP activities indicate that these enzymes do not contribute to alterations in acetyl-CoA metabolism in TD N9 microglial cells. On the contrary, the observed strong inhibition of those enzymes in cholinergic neuronal cells indicates their high vulnerability to neurotoxic conditions (Figure 2d, Figure 4c,d and Figure 8) [38].

Thus, our studies revealed an uneven distribution of ThDP in neuronal, microglial and astroglial cells with its highest level in N9 microglia (153 pmol/mg of protein), intermediate in SN56 neurons (95 pmol/mg of protein) and the lowest level in C6 astroglial cells (72 pmol/mg of protein) (Figure 1a, Table 1) [8]. They, however, did not correlate with cell susceptibility to TD, the highest one being for SN56, intermediate for N9 microglia and the lowest level for astroglia C6 (Figure 1b,d and Figure 3c,d, Table 1) [8]. This particular susceptibility of SN56 cells to TD precisely results from their high demand for acetyl-CoA (Figure 8). These findings remain in accord with human and animal studies revealing that the dysfunction of cholinergic transmission is a primary cause of cognitive and motor disturbances in TD subjects [14]. An excessive accumulation of Zn in the synaptic cleft of extensively depolarized glutamatergic nerve terminals is known to exert cytotoxic effects on postsynaptic neurons and cultured cholinergic neuronal cells [8,9,22,27]. The toxicity of this cation appears when its concentration exceeds the binding capacity of proteins present in the extracellular compartment [25]. For the medium containing a 10% FBS, such a saturating borderline Zn was estimated to be about 0.10 mmol/L [26,27]. However, in the brain, the extracellular fluid protein levels are about 10 times lower, whereas Zn levels may be in the range of 0.1–0.3 mmol/L, during a sub-second time of depolarization [25]. It indicates that the extracellular Zn used in this study corresponds to the low level of the biologically active free cation in vivo, which is able to penetrate plasma membranes and accumulate in neurons and microglial and astroglial cells. As a consequence, its intracellular concentration was markedly increased irrespective of ThDP status (Figure 3b, Table 1) [8]. These values correspond to the brain’s Zn levels accumulating in postsynaptic neurons and peri-synaptic glial cells during the TD-evoked stimulation of glutamatergic terminals [25]. However, such levels of accumulating Zn did not affect the viability and metabolic parameters in mildly ThDP-deficient glial N9 and C6 cells (Figure 1b,d, Table 1) [8]. On the other hand, thiamine-saturated cholinergic neurons tolerated a slight Zn excess, but they were highly sensitive to these conditions being in mild TD (Table 1). N9 microglia, due to the slow rate of energy metabolism in resting conditions, presented moderate responses to TD or Zn overload, in the form of smaller decreases in acetyl-CoA levels and aconitase and ICDH activities than in SN56, in the same conditions (Figure 3, Figure 4, Figure 7a–d and Figure 8, Table 1).

Thiamine deficiency in the brain is frequently combined with the excitotoxic activation of glutamatergic neurons [1]. Our past data revealed that the PDHC is a common target for the cytotoxic effects of Zn overload and ThDP deficits in the brain in vivo. Zn would inhibit the E2 subunit by lipoamide removal, whereas amprolium would inhibit the E1 subunit by ThDP depletion from its active center [26]. Therefore, the borderline cytotoxic effects of marginal Zn excess may be significantly aggravated by coexisting ThDP deficits in cholinergic neuronal cells (Figure 1, Figure 2, Figure 3, Figure 4 and Figure 7d–f).

The strongest, complete inhibition, which was observed for the PDHC in SN56 neurons grown in a medium containing Zn and amprolium, may significantly impair acetyl-CoA provision for mitochondrial energy synthesis and cytoplasmic cholinergic metabolism. Similar conditions could also suppress OGDHC activity, being a rate-limiting step for the second part of the TCA cycle [14]. Such mechanisms would be compatible with clinical and experimental observations, which reveal worsening ACh-dependent cognitive and motor functions in thiamine-deficient AD or alcoholic patients, as well as in AD-transgenic animals (Figure 3c,d and Figure 4) [2]. It is known that under diverse in vivo and in vitro neuro-inflammatory conditions, microglia exert cytotoxic effects against neuronal cells by boosting the release of numerous cytokines and thromboxane derivatives [1]. Moreover, neurons themselves synthesize and release limited amounts of these compounds, which were reported to be detrimental to them [42]. However, in this study, the inserted N9 microglia exerted partial protective effects against TD SN56 neuronal cells heavily impaired by Zn overload (Figure 3 and Figure 4). These beneficial effects were caused by the restoration of the control level of intraneuronal Zn and the partial alleviation of TD (Figure 3a,b). This partially restored PDHC activity yielded increases in acetyl-CoA levels and aconitase and ICDH-NADP activities and thereby energy production (Figure 4) [32]. Nevertheless, these positive alterations in SN56 metabolism brought about only a small restoration of their growth potential, despite the marked elimination of dysfunctional damaged cells (trypan blue positive cells) (Figure 3c,d). Thus, the relatively weak neuroprotection could be due to the stimulation of cytokine production by SN56 or N9 cells [43]. However, the concentration of IL-6 in the medium was not changed irrespectively in the tested conditions (Figure 6a). In addition, the bulk of IL-6 originated from neuronal cells, as N9 inserts caused no change in its level (Figure 6a). On the other hand, 90% of TNF-α originated from microglial cells as the N9 insert caused a 10-fold increase in this cytokine in the thiamine-deficient medium (Figure 6b). However, the concentration of 2 nmol/L appeared to be not neurotoxic as no changes in the metabolic and vitality parameters were observed in these conditions (Figure 3, Figure 4 and Figure 6b). Moreover, the TNF-α level was decreased to a non-measurable level in most neurotoxic conditions (Figure 3 and Figure 6b). These findings exclude the participation of both cytokines in the neurotoxicity or neuroprotection mechanism in these conditions, despite the fact that their concentrations in the media were higher or equal to specific receptor binding constants varying from 0.5 to 5.0 nmol/L [44,45]. One should stress that medium IL-6 and TNF-α concentrations are within the 0.1–3.0 nmol/L range which is compatible with their possible binding to TNF-α or IL-6 receptors [46]. On the other hand, they remain in accord with reports on the neuroprotective properties of the M2 microglia phenotype in neuroinflammatory conditions [38,42]. In addition, the suppression of TNF-α by Zn or amprolium could constitute a kind of auto-protective mechanism (Figure 7b). 

Our data suggest that the neuroprotection of N9 against the TD-affected Zn excitotoxicity of SN56 cholinergic neurons may be linked with the uptake of the ion excess by N9 cells. They accumulate Zn excess competitively to SN56 resulting in a decrease in the cation concentration in the former both in control and severe ThDP deficiency conditions Figure 4b and Figure 8). However, these effects of N9 were weaker than those exerted by the same number of C6 astroglial cells (see Methods Section 4.1) [8]. It may result from the two times higher ability of astroglial cells to accumulate Zn and their higher resistance to this cation excess (Figure 7a,f). These data indicate that microglia located in the peri-synaptic region could support astroglia in the tripartite synapse complex helping in clearing Zn excess from the overstimulated glutamatergic synapse and alleviating excitotoxic conditions in the diseased brain. Thereby, they could help restore PDHC activity and acetyl-CoA supply for energy production and ACh synthesis yielding an improvement in brain cognitive functions. Such a mechanism should be particularly effective in the mild disturbances of energy metabolism taking place in the early stages of cognitive disorders.

One should take into account that, except for the mitochondrial mechanisms presented here, there are chronic thiamine-dependent disturbances in citrate metabolism, which, through ATP-citrate lyase, provides acetyl-CoA for cytoplasmic ACh synthesis and microsomal and nuclear acetylations [7].

## 4. Materials and Methods

### 4.1. Cell Cultures 

The cellular model of Zn excess in TD was microglial N9 cell line (RRID: CVCL 0452) and cholinergic neuroblastoma SN56.B5G4 cells (RRID: CVCL 4456). To achieve more cholinergic phenotype, the cells were differentiated by combined addition of 1 mmol/L dibutyryl cAMP (cAMP) and 0.001 mmol/L all-trans-retinoic acid (RA) for 48 h [47]. At this time, the medium was replaced in both groups with fresh media, without differentiating factors. The cultures were continued for the next 24 h. Both lines were checked for their phenotype once a month. SN56 cells were verified by acetylcholine level measurement, which should be no less than 30 pmols/mg of protein in highly differentiated cholinergic cells. N9 cells were verified by the aggravation of Il-6 release by LPS (Figure 5). Cells of both lines grown in Minimal Eagle Medium, (MEM), containing 1 mmol/L L-glutamine, 0.05 mg of streptomycin and 50 U of penicillin per 1 mL and 10% fetal bovine serum (endogenous Zn, 0.005 mmol/L and 0.003 mmol/L thiamine). The number of passages before each experiment was 2 for both cell lines. Cells were seeded on 10 cm diameter plates at a density of 40.000/cm^2^. The culture was kept at 37 °C in an atmosphere of 5% CO_2_ and 95% air for 48 h. At this time, the growth medium was replaced with fresh one, containing Zn, amprolium or both, as indicated. The culture was continued for the next 24 h. 

The control cells were cultured in MEM supplemented with 0.009 mmo/L of thiamine concentration. 

Cultures were terminated by harvesting the cells in 10 mL of ice-cold 154 mmol/L NaCl containing 5 mmol/L glucose and 10 mmol/L phosphate buffer (pH 7.4). Then, the cells were collected and centrifuged (10 rpm for 7 min), and the cellular pellet was suspended in 0.20 mL of 320 mmol/L sucrose with 0.10 mmol/L EDTA and 10 mmol/L Na-HEPES (pH 7.4). Each sample was aliquoted for trypan blue assay and metabolite and enzyme assays. 

To determine the combined effect of N9 cells on Zn neurotoxicity, SN56 cells were co-cultured with N9 cells. The microglial N9 cells (3 × 10^5^) were seeded on 3 cm diameter inserts with a semipermeable bottom. After 24 h of preconditioning, 3 inserts were placed on each 10 cm plate with SN56 cells. The whole culture was continued for the next 24 h. Then, amprolium, Zn or both were added as indicated, and co-culture was continued for the subsequent 24 h.

Immediately after harvesting, cells were counted and viability tested with trypan blue exclusion assay. PDHC suspensions were frozen at −20 °C and used within 5 days. Acetyl-CoA cells were immediately incubated in depolarizing medium for 30 min and deproteinized as described below (see Section 4.4). The levels of intracellular Zn and ThDP were assessed after washing cells with Ca-free incubation medium containing 1 mM EDTA and deproteinization with ice-cold 4% (*w*/*v*) HClO_4_ for Zn assay (see Section 2.4) and 0.5% trichloroacetic acid (TCA) for ThDP assay (see Section 4.5).

### 4.2. Trypan Blue Exclusion Assay

Immediately after harvesting the cells, 0.02 mL of cell suspension was mixed with equal volume of 0.4% (*w*/*v*) isotonic trypan blue solution. Total cell number and fraction of nonviable, dye-accumulating cells (trypan blue positive cells) were counted after 2 min in Fuchs-Rosenthal haemocytometer under light microscope.

### 4.3. Enzyme Assays

Before the assay, samples were diluted to desired protein concentration in 0.20% Triton X-100. PDHC activity was assayed by the citrate synthase coupled method [48]. Enzyme activity was estimated by the two-step citrate synthase coupled method. The measurement of synthesized acetyl-CoA followed by citrate measurement was determined. The 0.25 mL of incubation media consisted of (in mmol/L) 50 Tris/HCl pH = 8.3, 2 MgCl_2_, 10 dithiotreitol, 2 NAD, pyruvic acid, 0.20 CoA, 2.5 oxaloacetic acid, 0.15 IU citrate synthase and 2 thiamine pyrophosphate and 0.05 mL of tested sample containing 0.10 mg of protein, but no thiamine pyrophosphate was added to the incubation medium to mimic ThDP deficient environment. Next, the cells were incubated with media for 30 min at 37 °C. The incubation was terminated by putting the sample at 99 °C for 10 min. Then, the level of citrate was measured. The media for citrate measurement at the final volume of 0.70 mL consisted of (in mmol/L) 100 Tris/HCl buffer pH = 7.4, 0.10 NADH, 0.2 IU malic dehydrogenase and 0.20 mL tested sample. The reaction was started by the addition of 0.1 IU citrate lyase to the tested sample. The activity of the enzyme was presented as nmols of oxidized NADH/min/mg of protein, on the basis of an absorbance coefficient for NADH = 6.22 mol/cm at 340 nm.

Aconitase (EC 4.2.1.3) and IDHC-NADP (EC 1.1.1.42) activities were measured by a direct measurement of NADP reduction [49,50]. The incubation media for aconitase at the final volume of 0.80 mL consisted of (in mmol/L) 50 Tris/HCl pH = 7.4, 2 MgCl_2_, 0.10 NADP, 1 IU. isocitrate dehydrogenase and 0.10 mL of tested sample containing 0.10 mg of protein. The reaction was initiated by the addition of 0.01 mL 20 mmol/L of cis-aconitate. The activity of the enzyme was presented as nmols of reduced NADP on the basis of an absorbance coefficient for NADPH = 6.22 mol/cm.

The incubation media for ICDH-NADP at the final volume of 0.70 mL consisted of (in mmol/L) 50 Tris/HCl pH = 7.4, 0.60 MgCl_2_, 0.50 NADP and 0.10 mL of tested sample containing 0.10 mg of protein. The reaction was initiated by the addition of 0.01 mL 10 mmol/L L-isocitrate. The activity of the enzyme was presented as nmols of reduced NADP on the basis of an absorbance coefficient for NADPH = 6.22 mol/cm at 340 nm.

### 4.4. Metabolic Studies

Metabolic parameters of the cells concerning their acetyl-CoA were assessed by their incubation in the depolarizing medium containing the following in a final volume of 1.0 mL: 2.5 mmol/L pyruvate, 2.5 mmol/L malate, 90 mmol/L NaCl, 30 mmol/L KCl, 20 mmol/L NaHEPES (pH 7.4), 1.5 mmol/L Na-phosphate, 0.320 mmol/L sucrose and 0.7–1.0 mg of cell protein. These conditions were employed to normalize the metabolic flux in every sample [51]. Incubation was started by the addition of cell suspension and continued for 30 min at 37 °C with shaking at 100 cycles per min. Acetyl-CoA level was estimated by the cycling method. The primary incubated cells were deproteinized with 0.20 mL 4% HClO_4_. Deproteinized extracts were put into a solution containing maleic anhydride in ethyl ether to remove CoA-SH for 2 h. The cycling reaction was carried out in 0.10 mL of medium containing (in mmol/L) 1.9 acetyl phosphate, 1.2 oxalo-acetate, 1.0 IU phosphotransacetylase and 0.12 IU citrate synthase for 90 min. The reaction was terminated by putting the samples at 99 °C for 10 min. Then, the level of produced citrate was determined. The media for citrate measurement at the final volume of 0.70 mL consisted of (in mmol/L) 100 Tris/HCl buffer pH = 7.4, 0.10 NADH, 0.2 IU. malic dehydrogenase and 0.20 mL tested sample. The reaction was started by the addition of 0.1 IU citrate lyase to the tested sample. The activity of the enzyme was presented in mmols of oxidized NADH on the basis of an absorbance ratio for NADH = 6.22 mol/cm at the wavelength = 340 nm. 

The whole-cell acetyl-CoA level is a marker of energy metabolism integrity in brain cells [22].

### 4.5. Thiamine Pyrophosphate Assay

For ThDP assay, the cells were homogenized in ice-cold 4% TCA. ThDP was measured in supernatant by reverse-phase HPLC with electrochemical detector.

For ThDP assessment, cells were cultured on 3 cm diameter plates and treated as described above. After 72 h, media were withdrawn, and cells that were attached to the plates were gently washed once with ice-cold PBS supplemented with 1 mmol/L EDTA. Then, 1.5 mL of 0.5% trichloroacetic acid was added to the cells and incubated on ice for 1 h. Protein precipitate was removed by centrifugation at 5000 g for 60 s. Supernatant was used for ThDP measurement with use of the reverse-phase HPLC [52]. 

### 4.6. Protein Assay

Protein level was measured using Bradford method (1976) with human immunoglobulin as standard. 

### 4.7. Zinc Assay

Zinc level was determined using modified fluorimetric method with N-(6-methoxy-8-quinolyl)-p-toluenesulfonamide (TSQ) as described by Chen and Liao [53]. Culture was terminated by medium withdrawal. Cells that were attached to the plates were gently washed with ice-cold PBS, supplemented with 1 mmol/L EDTA to remove surface-bound metals. Next, 2 mL 4% HClO_4_ was added to the plate and incubated on ice for 1–2 min. The cells were collected into ice-cold plastic tubes. Protein precipitates were removed by centrifugation for 1 min at 10,000 rpm. The supernatants were neutralized prior to the assay with 3.25 M K_2_CO_3_ to pH 6.0. Next, the supernatants were used for Zn assay with modified fluorometric method TSQ with emission and excitation wavelengths equal to 335 and 495 nm, respectively, against a blank reagent that was treated similarly.

To quantify analysis, the peak of emission was calculated from calibration curve in range 0.2–4.0 nmols/sample of ZnCl_2_ standard solution treated similar to the samples [8].

### 4.8. Il-6 and TNF-α Assay

The level of cytokines was assayed using commercial Platinum ELISA kits (eBioscience, Bender MedSystems GmbH, Vienna, Austria). The two groups were compared by non-parametric Mann-Whitney test, at *p* < 0.05 being considered to be statistically significant.

### 4.9. Statistical Analyses

Statistical analyses were carried out using GraphPad Prism 8.0 (GraphPad software, La Jolla, CA, USA). Data are presented as means ± SEM from 3–6 independent cell culture preparations. The assessment of normality of the data were carried out by Shapiro-Wilk normality test. Most data were carried out by Kruskal–Wallis test. The two groups were compared by non-parametric Mann-Whitney test, at *p* < 0.05 being considered to be statistically significant.

## 5. Conclusions

The presented data indicate that in the brain, the combination of marginal ThDP deficit and borderline Zn excess may result in overt cytotoxic effects yielding a differential impairment of neuronal, microglial and astroglial cells (Figure 8). The differential susceptibility of neuronal and glial cells may depend on the individual ratios between their energy demands and the degree of inhibition of PDHC and acetyl-CoA production in these conditions (Figure 8, Table 1). Hence, thiamine supplementation can make any brain cell more resistant to Zn overdose, due to a decrease in their plasma membrane permeability. This study indicates that otherwise non-harmful levels of pathogenic signals in combination with others may reciprocally potentiate their toxicities focusing on crucial targets such as the PDHC, which is the principal source of acetyl-CoA, a key energy and structural substrate in the brain. Its shortages lead to diverse brain pathologies. Neuronal cells are more susceptible to ThDP deficits and Zn excess than microglial and astroglial cells due to their particular sensitivity of the PDHC to these inhibitors and a relatively high demand of acetyl-CoA for energy, N-acetyl-aspartate and neurotransmitter synthesis (Figure 8). Such neuro-glial compartmentation of acetyl-CoA and energy metabolism might point out the mechanism of early cognitive deficits appearing in diverse neuropathological conditions.

## Figures and Tables

**Figure 1 ijms-24-04465-f001:**
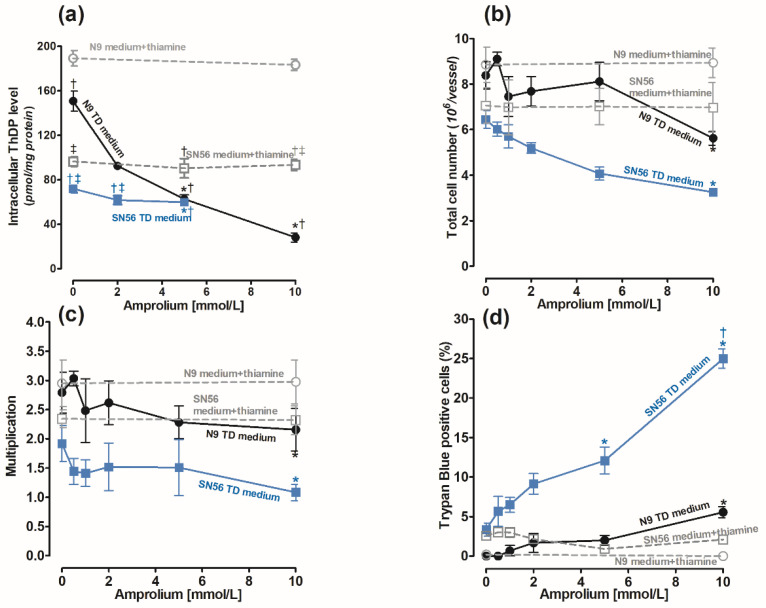
Concentration-dependent effects of amprolium on (**a**) ThDP content; (**b**) final cell counts; (**c**) growth rate calculated as ratio of final cell count/initial cell count; (**d**) loss of cell viability as trypan blue positive cell fraction in microglial N9 (●) and cholinergic neuronal SN56 (■) cells cultivated in TD medium and with 0.009 mmol/L thiamine addition. Data are means ± SEM from 5–6 independent cell-culture preparations. Significantly different from TD medium control * *p* < 0.01; corresponding thiamine-supplemented medium ^†^
*p* < 0.01; corresponding N9 medium ^‡^
*p* <0.01.

**Figure 2 ijms-24-04465-f002:**
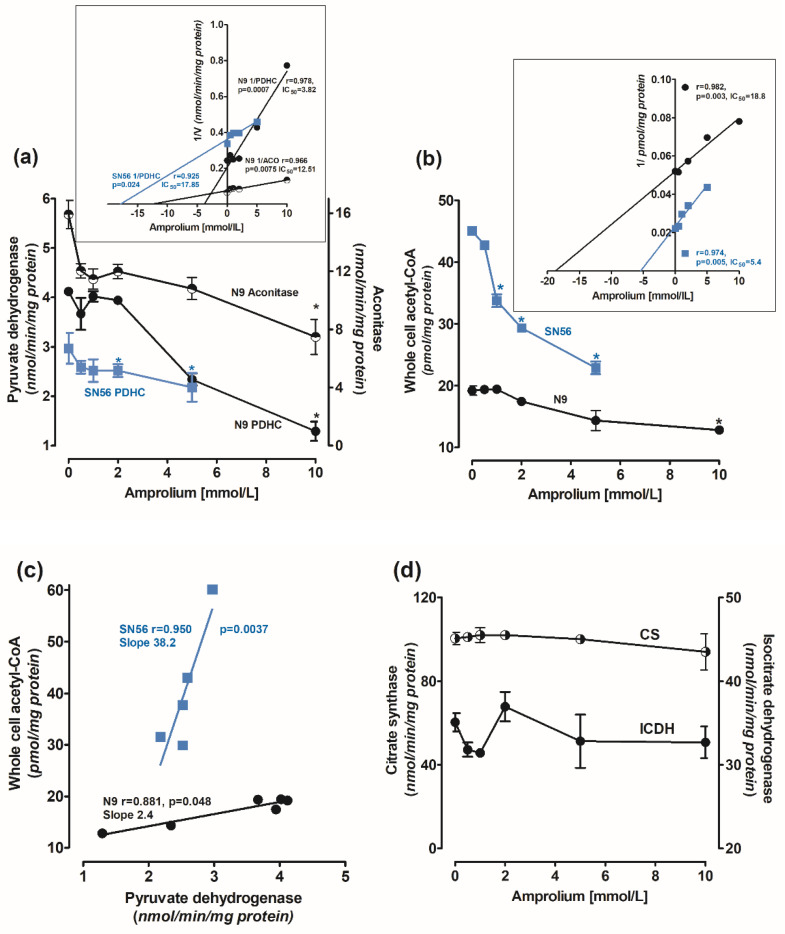
Concentration-dependent effects of amprolium on activities of (**a**) PDHC (measured in ThDP free assay medium) and aconitase; (**b**) acetyl–CoA levels; (**c**) correlation plots of acetyl–CoA vs. PDHC activity; and (**d**) IDH–NADP and citrate synthase activities in N9 microglial (●) and cholinergic neuronal SN56 (■) cells cultured in TD medium and with 0.009 mmol/L thiamine; (insert (**a**)) Dixon’s plots of concentration-dependent effects of amprolium on activities of PDHC and aconitase; (insert (**b**)) Dixon’s plots of concentration-dependent effects of amprolium on activities of PDHC and acetyl-CoA content. Data are means ± SEM from 4–5 independent cell-culture preparations. Significantly different from respective no-amprolium controls * *p* < 0.05.

**Figure 3 ijms-24-04465-f003:**
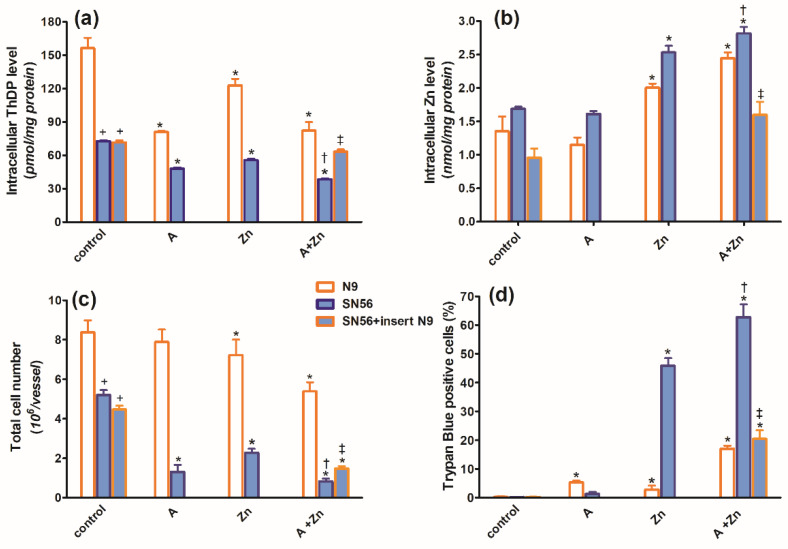
Chronic effects of 5 mmol/L (A) amprolium, 0.10 mmol/L Zn, A + Zn and N9 inserts on (**a**) intracellular ThDP level, (**b**) intracellular Zn level, (**c**) total cell number and (**d**) viability of N9 and SN56 cells cultured in TD conditions. Data are means ± SEM from 4–6 independent cell-culture preparations. Significantly different from, respectively, control * *p* < 0.01; amprolium alone ^†^
*p* < 0.05; Zn alone ^‡^
*p* < 0.05; N9 control ^+^
*p* < 0.01.

**Figure 4 ijms-24-04465-f004:**
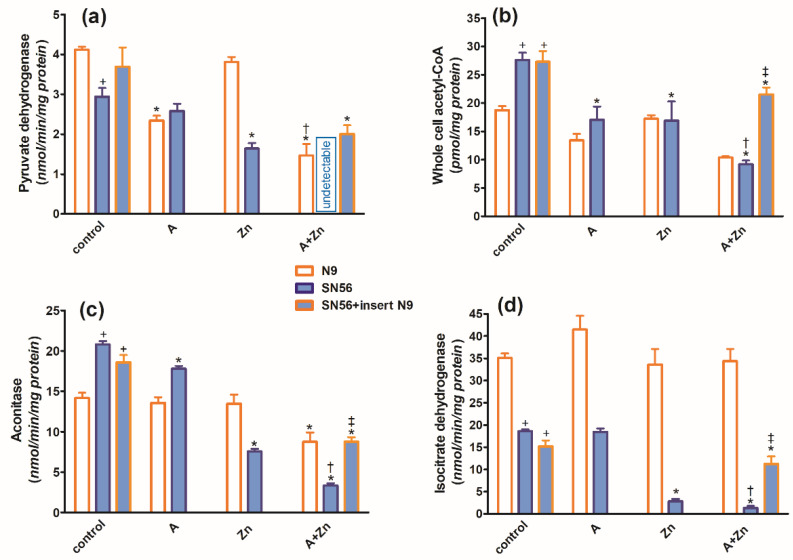
Chronic effects of 5 mmol/L (A) amprolium, 0.10 mmol/L Zn, A + Zn and N9 inserts on (**a**) PDHC activity, (**b**) acetyl-CoA level, (**c**) aconitase activity and (**d**) ICDH-NADP activity in cells cultured in TD conditions. Data are means ± SEM from n = 4–6 independent cell-culture preparations. Significantly different from, respectively, control * *p* < 0.01; amprolium ^†^
*p* < 0.05; Zn ^‡^
*p* < 0.05; N9 control **^+^**
*p* < 0.01.

**Figure 5 ijms-24-04465-f005:**
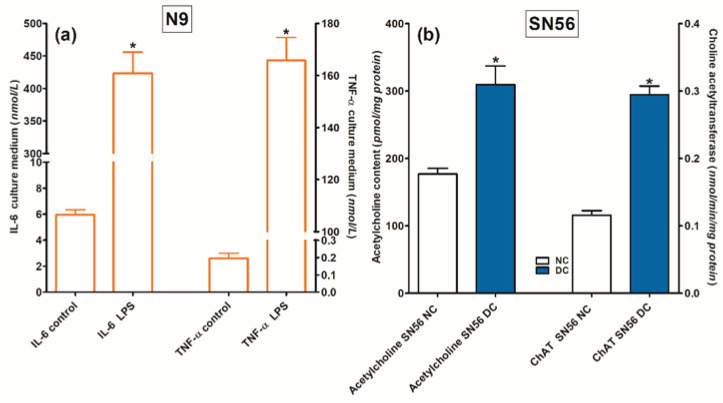
Functional capacity of (**a**) microglial N9 cells tested by stimulating effects of LPS on IL-6 and TNF-α release to the growth medium and (**b**) cholinergic SN56 neuronal cells measured by cAMP-retinoic acid differentiation-evoked increases in ACh content and ChAT activity. Significantly different from respective control * *p* < 0.01.

**Figure 6 ijms-24-04465-f006:**
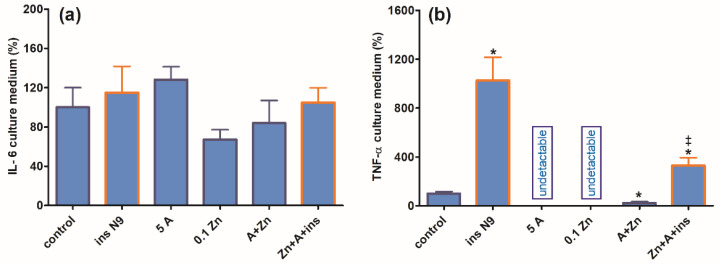
Chronic effects of 5 mmol/L (A) amprolium and 0.10 mmol/L Zn, A + Zn and N9 inserts on (**a**) IL-6 and (**b**) TNF-α released from SN56 neuronal cells. Data are means ± SEM from 4–6 independent cell-culture preparations. Significantly different from, respectively, control * *p* < 0.01; from A + Zn and ins.N9 ^‡^
*p* < 0.01.

**Figure 7 ijms-24-04465-f007:**
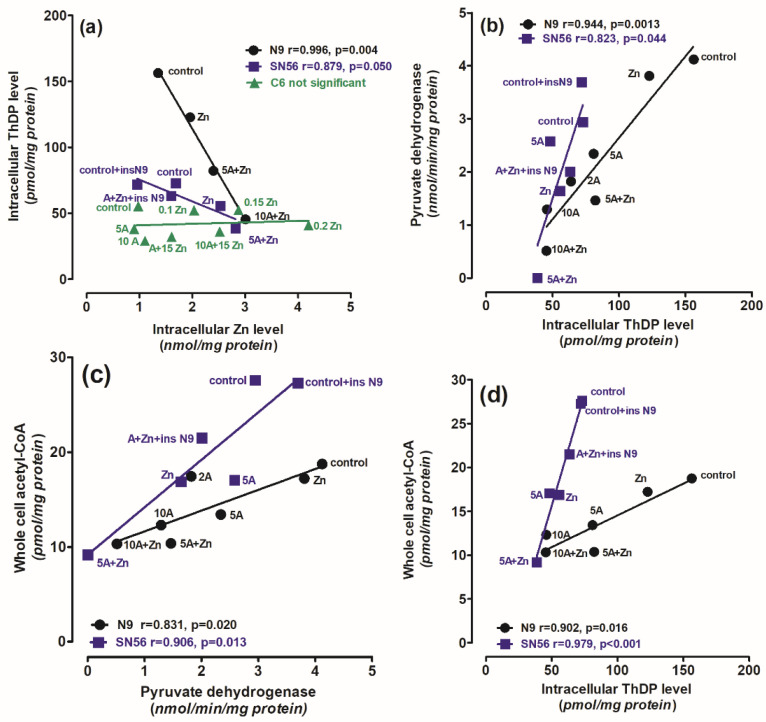
The correlations between (**a**) intracellular Zn level and intracellular ThDP level; (**b**) PDHC activity assayed in medium without ThDP and intracellular Zn level; (**c**) whole-cell acetyl-CoA level and PDHC activity assayed in medium without ThDP; (**d**) whole-cell acetyl-CoA level and intracellular ThDP level; (**e**) final total cell number and whole-cell acetyl-CoA level; (**f**) nonviable trypan blue positive cells and intracellular Zn level in N9 microglial (●) and SN56 cholinergic (■) neuronal cells (cultured in thiamine-deficient media). Data were taken from Figure 1, Figure 2, Figure 3 and Figure 4 and [32] (adapted with permission from [32]).

**Figure 8 ijms-24-04465-f008:**
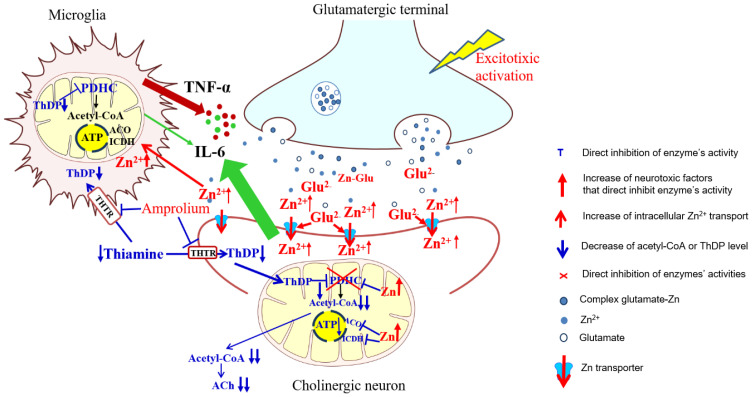
Summary of cytotoxic and protective interactions between cholinergic neuronal and microglial cells in neuro-microglial tripartite synapse model. Different harmful signals may activate glutamatergic neurons to release excess of glutamate and Zn from their terminals, which may be cleared from the synaptic cleft by postsynaptic cholinergic neurons and microglial cells. Glutamate activating NMDA channels facilitates Zn entry into the cells. Zn overload inhibits PDHC complex and aconitase (ACO) and ICDH-NADP activities, impairing both acetyl-CoA synthesis and its utilization in TCA cycle, yielding deep deficits of acetyl-CoA for ACh synthesis in cytoplasm and ATP production in mitochondria, respectively. Thiamine enters cells through THTR transporters, which are competitively inhibited by amprolium [41]. In cytoplasm, thiamine is phosphorylated by respective kinases to ThDP. TD aggravates Zn-evoked alterations by co-inhibition of PDHC resulting in deep shortages of energy and death of cholinergic neurons. Adjacent microglia may partially alleviate these symptoms by taking up part of Zn from the synaptic cleft thereby decreasing Zn load in neurons. Marginal Zn excess in microglia inhibits none of these enzymes. Only PDHC displays susceptibility to ThDP deficits. This makes microglial cells more resistant to excitotoxic and TD conditions. These data indicate that, besides astroglia, microglia might also display neuroprotective properties, under specific conditions. Legend: Red letters and arrows indicate increase or activation. Blue letters and dashed and sharp arrows indicate inhibition or decrease.

**Table 1 ijms-24-04465-t001:** Chronic effects of Zn in thiamine-supplemented and thiamine-deficient medium on metabolic, functional and viability parameters of N9 and SN56 cells.

Parameter	N9 Microglial Cells		SN56 Cholinergic Neuronal Cells		C6 Astroglial Cells [16]	
	MEM + Thiamine	MEM	MEM + Thiamine	MEM	MEM + Thiamine	MEM
	Relative Values (Percent of Control)					
	Zn 0.10		Zn 0.10		Zn 0.10	
TB (+) (% of total)	0.1 ± 0.02	3 ± 0.01 *	1 ± 0.07	47.0 ± 0.5 ***^†‡^**	n.d.	n.d.
Cell counts	106 ± 2	114 ± 30	102 ± 4	44 ± 10 ***^†‡^**	105 ± 15	106 ± 2.0
Proliferation rate	2.6 ± 0.1 Control: 2.9 ± 0.2	2.8 ± 0.2 Control: 2.8 ± 0.6	2.4 ± 0.1 Control: 2.5 ± 0.1	0.4 ± 0.1 ***^†^** Control: 1.7 ± 0.2	2.1 ± 0.2 Control: 1.95 ± 0.3	2.2 ± 0.1 Control: 2.08 ± 0.3
Zn intracellular	154 ± 6 *	147 ± 11 *	150 ± 3 *	149 ± 9 *	150 ± 12	208 ± 4 ***^†‡^**
ThDP level	nd.	78 ± 6 *	98 ± 7	75 ± 12 *	n.d	87 ± 3
PDHC (no ThDP)	101 ± 10	92 ± 15	106 ± 11	44 ± 7 ***^†‡^**	107 ± 19	88 ± 4
Acetyl-CoA	149 ± 14 *	92 ± 9	107 ± 12	80 ± 5 ***^†^**	89 ± 5	93 ± 5
Aconitase	81 ± 6	95 ± 11	99 ± 12	33 ± 5 ***^†‡^**	95 ± 4	113 ± 7
ICDH-NADP	106 ± 10	105 ± 12	96 ± 4	15 ± 6 ***^†‡^**	105 ± 14	96 ± 9

Part of the data for SN56 cells were taken from [8,9] (adapted with permission from [8,9]) and Figure 1 and Figure 2; for N9 cells from Figure 1 and Figure 2; for C6 cells from [8]. Thiamine was added at the 0.009 mmol/L concentration. Significantly different from, respectively, no Zn control, * *p* < 0.05; control; +thiamine, **^†^**
*p* < 0.05; N9 or C6, **^‡^**
*p* < 0.05. Abbreviations: nd., not determined. Proliferation rate is a ratio of harvested cell count to count of seeded cells; values of controls represent proliferation rate of cells in the absence of Zn in the medium.

## Data Availability

The data that support the findings of this study are available from the corresponding author upon reasonable request.

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
