# Peer review of "Effects of Marginal Zn Excess and Thiamine Deficiency on Microglial N9 Cell Metabolism and Their Interactions with Septal SN56 Cholinergic Cells"

_ijms, 2023, doi:10.3390/ijms24054465_

Round 1

Reviewer 1 Report

1. This is kind of research needs to repeat at least two times.

2. Introduction needs to get shorter and just related references cited.

3. Despite presenting such an important topic, the authors failed to provide a clear description of the methodology. There is no systematic material and methods in the manuscript. Immediately after introduction, Results have written which is not routine way in writing of scientific manuscript. First you need to write material and methods with details. 

4. References need to sort in one format based on MDPI publication format. In some references year located at the end like ref. N.S,R,B but in the others in different way! 

5. The information of results was a bit lost in the Abstract. 

6. The discussion is vague, full of speculations, with the mention of just few papers on the matter. There is no cohesion and no fluidity of the text.

Author Response

The answers to the Rewiever I comments:

  1. This is kind of research needs to repeat at least two times.

Data are means from  4 -7 in all experiments.

  1. Introduction needs to get shorter and just related references cited.

The introduction is shortened.

  1. Despite presenting such an important topic, the authors failed to provide a clear description of the methodology. There is no systematic material and methods in the manuscript. Immediately after introduction, Results have written which is not routine way in writing of scientific manuscript. First you need to write material and methods with details. 

The methods are rewritten and more details are given with dedicated references. The Materials and Method chapter is at the end of the manuscript because we organized it along with indications of the template provided by the Editorial Office. So we had no choice. All articles in this journal are published in accord with this scheme. Honestly, we prefer having materials and methods section first. We remain it to decision of Editor.

  1. References need to sort in one format based on MDPI publication format. In some references year located at the end like ref. N.S,R,B but in the others in different way! 

The references are corrected

  1. The information of results was a bit lost in the Abstract. 

The results are given at the Abstract.

  1. The discussion is vague, full of speculations, with the mention of just few papers on the matter. There is no cohesion and no fluidity of the text.

The discussion is edited.

Reviewer 2 Report

The manuscript by Anna Ronowska and co-authors describes the experiments which are very interesting for this field of metabolomics and cellular biochemistry. The work can be considered as a continuation of the previous paper, published in IJMS (https://doi.org/10.3390/ijms222413337). The problem is, this new manuscript is still a draft, due to the bad quality of description of Methods and the number of typos and mistakes in the text. Together these factors make reading and evaluation of the data very difficult and raises multiple questions to the reliability. For example, the "Results" section mentions the presence of thiamine in the "thiamine deficient" medium, which contradicts with the methods. Clear understanding of this is very important for this manuscript.

Generally a careful rewriting of the introduction and abstract is also needed, together with the addition of proper citations.

The style of the figures can be improved, together with the addition of an explanatory schemes too. It would be a pity if there are typos such as in the Fig. 1(a) of the previous paper cited above ("Thiamine pyrophospate").

Some other comments and questions are listed below. However, since the required revision should take much time and efforts, I believe, the authors should resubmit the manuscript, when it's truly ready. It would be a pleasure for me to read it again after a significant and careful improvement.

Some comments and questions (Q) to the current text:

TDP - is used for thymidine diphosphate. It's better to use ThDP or TPP instead.

"Energy deficits are seen in brains of patients with several neurodegenerative diseases as well as in animal models of encephalopathies" (line 30) - Please be more precise and name the diseases and/or models. Such sentence may also require more references (different ones for different conditions/models).

"Inhibition of pyruvate dehydrogenase (PDHC), α-ketoglutarate dehydrogenase (KDHC) complexes in the brain may be the key factor in this pathomechanism." - (line 33) - This also needs a better reasoning, maybe references. In addition, it would be good to name OGDH gene at least once because of the problem with KDHC/OGDHC tradition in the literature. The latter is disclosed further in the introduction (10+ lines later). From the introduction it seems, the role of KDHC would be disclosed too, but only PDHC is studied of the two enzyme complexes.

Q1. This example (in the comment above) may reveal the overall problem of the introduction and abstract: there are right statements, but their combinations are not very logical and repeats are rather common. Not optimal use of abbreviations sometimes makes it worse. Complex sentences with missing prepositions also make the text difficult to read.

Generally, the introduction needs more accurate style of citing the references (e.g. lines 68-75 and other examples mentioned above).

Lines 84-86 - rephrase the sentence (... "brought about"...).

Q2. Such a work and description of rather high Zn concentrations in the introduction must be balanced by mentioning of regular Zn concentrations inside the cell and in blood plasma and CSF (mitochondrial concentrations would be also great to mention because of the focus on PDHC and the TCA cycle).

Q3. Since 0.003 mmol/L thiamine is present in the "thiamine deficient" MEM, it should be described as "low thiamine". This is completely different to what is stated in the abstract and text. Moreover, no information can be found in Methods (paragraph 4.1 and further). Thus, the model of TD should be clarified, since different models of TD exist, and there should be no mistakes or wrong interpretations here.

Q4. Chapter 4.2 has no references. What's more important, it doesn't describe all the corresponding protocols and includes a mistake. The chapter should be rewritten. Chapter 4.3 also should be written more carefully and should include a brief description of the assay. Chapter 4.4 is badly written. It's taken from another place and doesn't suit to this paper (e.g. TPP instead of TDP). How could anyone try to check the experiments after such a bad description of the methods? Other chapters of the "Methods" should be improved too.

Author Response

The answers to the Rewiever II comments:

  1. The manuscript by Anna Ronowska and co-authors describes the experiments which are very interesting for this field of metabolomics and cellular biochemistry. The work can be considered as a continuation of the previous paper, published in IJMS (https://doi.org/10.3390/ijms222413337). The problem is, this new manuscript is still a draft, due to the bad quality of description of Methods and the number of typos and mistakes in the text. Together these factors make reading and evaluation of the data very difficult and raises multiple questions to the reliability. For example, the "Results" section mentions the presence of thiamine in the "thiamine deficient" medium, which contradicts with the methods. Clear understanding of this is very important for this manuscript. Generally a careful rewriting of the introduction and abstract is also needed, together with the addition of proper citations. The style of the figures can be improved, together with the addition of an explanatory schemes too. It would be a pity if there are typos such as in the Fig. 1(a) of the previous paper cited above ("Thiamine pyrophospate").

The manuscript is rewritten, the Introduction chapter is shorter and more precise. The captions are corrected, the nomenclature of the media used in the study is uniformed. The Material and methods chapter is corrected, the details of the methods are given.

Some other comments and questions are listed below. However, since the required revision should take much time and efforts, I believe, the authors should resubmit the manuscript, when it's truly ready. It would be a pleasure for me to read it again after a significant and careful improvement.

 Some comments and questions (Q) to the current text:

  1. TDP - is used for thymidine diphosphate. It's better to use ThDP or TPP instead.

ThDP is used instead of TDP in all text.

  1. "Energy deficits are seen in brains of patients with several neurodegenerative diseases as well as in animal models of encephalopathies" (line 30) - Please be more precise and name the diseases and/or models. Such sentence may also require more references (different ones for different conditions/models).

These details are added to the text.

  1. "Inhibition of pyruvate dehydrogenase (PDHC), α-ketoglutarate dehydrogenase (KDHC) complexes in the brain may be the key factor in this pathomechanism." - (line 33) - This also needs a better reasoning, maybe references. In addition, it would be good to name OGDH gene at least once because of the problem with KDHC/OGDHC tradition in the literature. The latter is disclosed further in the introduction (10+ lines later). From the introduction it seems, the role of KDHC would be disclosed too, but only PDHC is studied of the two enzyme complexes.

OGDH is used instead of KDHC and the fragment explain its inhibition by cytotoxic factors is removed from the Introduction.

  1. This example (in the comment above) may reveal the overall problem of the introduction and abstract: there are right statements, but their combinations are not very logical and repeats are rather common. Not optimal use of abbreviations sometimes makes it worse. Complex sentences with missing prepositions also make the text difficult to read.

It is changed to be more fluid.

  1. Generally, the introduction needs more accurate style of citing the references (e.g. lines 68-75 and other examples mentioned above).

The references are added.

  1. Lines 84-86 - rephrase the sentence (... "brought about"...).

It is rephrased.

  1. Such a work and description of rather high Zn concentrations in the introduction must be balanced by mentioning of regular Zn concentrations inside the cell and in blood plasma and CSF (mitochondrial concentrations would be also great to mention because of the focus on PDHC and the TCA cycle).

The concentrations are given with proper references.

  1. Since 0.003 mmol/L thiamine is present in the "thiamine deficient" MEM, it should be described as "low thiamine". This is completely different to what is stated in the abstract and text. Moreover, no information can be found in Methods (paragraph 4.1 and further). Thus, the model of TD should be clarified, since different models of TD exist, and there should be no mistakes or wrong interpretations here.

It is corrected, thiamine deficient medium contains 0.003 mmol/L thiamine and can be used as a model of thiamine deficiency.

  1. Chapter 4.2 has no references. What's more important, it doesn't describe all the corresponding protocols and includes a mistake. The chapter should be rewritten. Chapter 4.3 also should be written more carefully and should include a brief description of the assay. Chapter 4.4 is badly written. It's taken from another place and doesn't suit to this paper (e.g. TPP instead of TDP). How could anyone try to check the experiments after such a bad description of the methods? Other chapters of the "Methods" should be improved too.

The Methods are rewritten and detail of the procedures are given with relevant references.

Round 2

Reviewer 1 Report

Thanks for revision. Now it is much better.

Author Response

Unfortunately, the text still contains a lot of typos, problems with abbreviations and English. The quality of the text should be improved, only then it could be published.

A couple of important comments should be incorporated as well.

Line 55. "ThDP is a cofactor of E1 subunits of PDHC and OGDHC." - and also BCDHC and OADHC. Not to say about TKTs and the ThDP-dependent alpha oxidation enzymes. Please correct.

We added recommended references

Line 99. After the description of Zn role here it would be great to add a link to thiamine (about thiamine release into the synaptic cleft, together with acetylcholine by the way, first shown by Minz in 1938 and confirmed further by other data).

We took care of thiamine as a substrate for ThDP synthesis by intracellular kinases. We interested in intracellular ThDP level only as cofactor for ThDP-dependent enzymes. Thiamine release if any was not important for us. It would be pure speculation to asses such turnover irrespective of 3 or 9 mmol/L thiamine is present in the medium. Nevertheless we included a statement which does not add to the rationale of this study.

Lines 111-114. The problem is, you don't give any references here. Generally the mentioned concentrations agree with the common concentrations used in DMEM, but many components of the DMEM composition are commonly considered to be too high. So, please support your statements by independent references here (not self citations I mean). And what would be of great help here, you can right here use the Fig 1, where you have significant differences, although they are not so impressive for just the effect of TD media vs control, but anyway.

We found nobody who checked thiamine level in MEM or DMEM. Some surplus of thiamine was apparently not harmful. In Fig.1 we measured thiamine diphosphate inside the cells and did not made assessments of extracellular thiamine in DMEM. Non proportional ThDPin accumulation vs thiamineex results from saturating character of the thiamine transport and thiamine kinase reactions.

Due to the importance of the point, I would insist on adding the info on thiamine concentration into abstract (line 15) as "Within this study we tested the effect of Zn on microglial cells cultivated in thiamine deficient medium (0.003 mmol/L vs 0.009 mmol/L in control medium)".

We added the information in the abstract.

You should also correct the mistake, because the  part "in thiamine deficient medium" is repeated twice in the abstract.

We removed the repetition

Many other typos could be found throughout the text. As well as the problems with abbreviations. A couple of examples can be found below.

Line 88 vs 55: OGDH or OGDHC? KDHC can be found in line 376.

Line 91: Zn2+ or Zn?

We changed Zn2+ to Zn, we corrected the other typos.

Reviewer 2 Report

The manuscript was significantly improved. The data are valuable and interesting. Unfortunately, the text still contains a lot of typos, problems with abbreviations and English. The quality of the text should be improved, only then it could be published.

A couple of important comments should be incorporated as well.

Line 55. "ThDP is a cofactor of E1 subunits of PDHC and OGDHC." - and also BCDHC and OADHC. Not to say about TKTs and the ThDP-dependent alpha oxidation enzymes. Please correct.

Line 99. After the description of Zn role here it would be great to add a link to thiamine (about thiamine release into the synaptic cleft, together with acetylcholine by the way, first shown by Minz in 1938 and confirmed further by other data).

Lines 111-114. The problem is, you don't give any references here. Generally the mentioned concentrations agree with the common concentrations used in DMEM, but many components of the DMEM composition are commonly considered to be too high. So, please support your statements by independent references here (not self citations I mean). And what would be of great help here, you can right here use the Fig 1, where you have significant differences, although they are not so impressive for just the effect of TD media vs control, but anyway.

Due to the importance of the point, I would insist on adding the info on thiamine concentration into abstract (line 15) as "Within this study we tested the effect of Zn on microglial cells cultivated in thiamine deficient medium (0.003 mmol/L vs 0.009 mmol/L in control medium)".

You should also correct the mistake, because the  part "in thiamine deficient medium" is repeated twice in the abstract.

Many other typos could be found throughout the text. As well as the problems with abbreviations. A couple of examples can be found below.

Line 88 vs 55: OGDH or OGDHC? KDHC can be found in line 376.

Line 91: Zn2+ or Zn?

Author Response

(The authors gave the same response as above.)
